

# Screening of MYB1R1 interaction with *LDOX* promoter to regulate anthocyanin biosynthesis in peaches

Xinxin Wu[1], Tong Du[1], Yan Li[1], Weibing Zhuang[2], Naixin Kang[1], Jiaxin Zeng[1,3], Cong Yan[1], Zhenzhu Hu[1] and Zewen Cao[4]

[1] School of Life Science and Food Engineering, Huaiyin Institute of Technology, Huai'an, Jiangsu, China
[2] Jiangsu Key Laboratory for the Research and Utilization of Plant Resources, Institute of Botany, Jiangsu Province and Chinese Academy of Sciences (Nanjing Botanical Garden Memorial Sun Yat-Sen), Nanjing, Jiangsu, China
[3] Institute of Pomology, Jiangsu Academy of Agricultural Sciences/Jiangsu Key Laboratory for Horticultural Crop Genetic Improvement, Nanjing, Jiangsu, China
[4] Beijing Huazhiyuan Horticulture Co., Ltd., Beijing, China

## ABSTRACT

**Background.** The floral color variegation of cultivar 'Sahong Tao' is distinctive and possesses significant ornamental value. Currently, there are no relevant reports on how MYB transcription factors (TFs) interact with *LDOX* promoter to regulate the flower color variegation in peach.

**Methods.** In this study, we screened for proteins that interact with the *LDOX* promoter using yeast one-hybrid (Y1H) and next-generation sequencing (NGS). The NGS data were aligned with the *Arabidopsis* database (TAIR10) utilizing Python 3.10.4. PlantTFDB was employed to identify TFs, while PlantRegMap was used to predict TFs that interact with the *LDOX* promoter. The Y1H assay verified MYB1R1 interaction with *LDOX* promoter, and Y1H-AOS predicted their binding sites. The physico-chemical properties, structure and interacting proteins of MYB1R1 were analyzed using bioinformatics methods. Sequence alignment and phylogenetic tree analyses of *MYB1R1* were performed. Finally, the tissue expression specificity of *MYB1R1* and *LDOX* in 'Sahong Tao' was examined using qRT-PCR.

**Results.** The Y1H and NGS results indicate that 1,190 proteins interact with the *LDOX* promoter. Among these, 20 TFs were identified, including ERF, MYB, NF-YB, SBP, S1Fa-like, TCP, bHLH, LBD, ZF-HD, C3H, DBB, MYB-related, and HD-ZIP. Of the 1,190 proteins, 1,146 exhibit high similarity to homologs in *Arabidopsis*, with 332 classified as RNA binding proteins and 124 as DNA binding proteins. A comparison with the NGS results identified seven TFs that align with predictions from PlantRegMap. Based on these findings, we selected MYB44 (PRUPE_6G229000, PRUPE_1G430000) and MYB1R1 (PRUPE_5G182000) as candidate members. Y1H assays demonstrated that MYB1R1 interacts with the *LDOX* promoter. Y1H-AOS was used to confirm 24 interaction binding sites. *MYB1R1* consists of an 897 bp full-length CDS, encoding 298 amino acids, with a predicted molecular weight of 32.49 kDa and a theoretical isoelectric point of 7.20. MYB1R1 features a typical SANT-MYB domain, and its secondary structure is predominantly composed of irregular coils. Phylogenetic analysis indicates a close evolutionary relationship between MYB1R1 from 'Sahong Tao' and both *Prunus avium* and *Prunus speciosa*. Promoter prediction analysis for *MYB1R1* reveals multiple hormone- and stress-related *cis*-acting elements. MYB1R1 may interact

Corresponding authors
Xinxin Wu, wxxhy@hyit.edu.cn
Weibing Zhuang, weibingzhuangnj@sina.com

with bHLH and other proteins to perform its functions. In variegated petals, *MYB1R1* expression is higher and *LDOX* expression is lower compared to red petals, suggesting that MYB1R1 negatively regulates anthocyanin synthesis by interacting with LDOX. This study contributes to elucidating the function of *MYB1R1* and the regulatory mechanism of MYB- *LDOX* in the flower color of 'Sahong Tao'.

## INTRODUCTION

Peach (*Prunus persica*), a small deciduous tree belonging to the genus *Prunus* in the *Rosaceae* family, exhibits a rich diversity of floral colors. Variegation is a distinctive floral color pattern, characterized by red-white bicoloration in the peach cultivar 'Sahong Tao' (*Chen et al., 2014*). This phenotype holds significant ornamental value, and is notably observed in other horticultural plants such as *Prunus mume, Rhododendron, Antirrhinum majus* and *Petunia hybrida* (*Nie et al., 2023*; *Piao, Wu & Cui, 2021*; *Wu et al., 2017*; *Yasumasa et al., 2012*). The variegated phenotype of 'Sahong Tao' is primarily associated with the types, concentration, and distribution of anthocyanins in petal cells (*Cheng et al., 2015*).

Anthocyanins are water-soluble flavonoid pigments (*Sun et al., 2016*) that impart a diverse range of colors to plant tissues and cells (*Khoo et al., 2017*). Previous studies of 'Sahong Tao' have demonstrated that red petals possess significantly higher concentrations of anthocyanins compared to variegated petals, suggesting that the red coloration primarily arises from the accumulation of anthocyanins (*Zhou et al., 2015b*). The formation of anthocyanins is regulated by a collaborative interaction between structural genes and transcription factors (TFs) (*Schwinn et al., 2016*). Structural genes are classified into two categories: early biosynthetic genes, which include *phenylalanine ammonia-lyase* (*PAL*), *cinnamate-4-hydroxylase* (*C4H*), *chalcone synthase* (*CHS*), and *chalcone isomerase* (*CHI*). Late biosynthetic genes including *dihydroflavonol-4-reductase* (*DFR*), *leucoanthocyanidin oxygenase/anthocyanidin synthase* (*LDOX/ANS*), and *UDP-glucose: flavonoid-3-O-glucosyltransferase* (*UFGT*). This classification is based on their specific roles in the biosynthetic pathway (*Chen et al., 2019*; *Sharma et al., 2024*). The primary role of LDOX is to facilitate the conversion of colorless anthocyanins into pigmented anthocyanins (*Owens & McIntosh, 2011*; *Zohar et al., 2015*). In *Arabidopsis thaliana*, a reduction in the allelic mutant of LDOX results in decreased levels of anthocyanins and proanthocyanidins, leading to a lighter seed coat coloration of the seed coat (*Bowerman et al., 2012*). In *Reaumuria trigyna*, two significantly differentially expressed genes, *RtLDOX* and *RtLDOX2*, have been identified. *RtLDOX2* enhances tolerance to abiotic stress by promoting the accumulation of anthocyanins and flavonols (*Li et al., 2021*), while the *RtLDOX* gene complements anthocyanidin synthesis (*Zhang et al., 2016*). In 'Sahong Tao', proteomic analyses have indicated that the protein abundance of LDOX is higher in red samples than in variegated samples (*Zhou et al., 2015a*). Further analysis demonstrates that

*LDOX* gene expression and enzyme activity are significantly greater in red samples (*Wu et al., 2020*). These findings suggest that *LDOX* plays a critical role in the formation of variegated flowers in 'Sahong Tao'. Currently, there is limited research on the interaction between MYB and the *LDOX* promoter in regulating anthocyanin synthesis in peach.

Various TFs participate in the anthocyanin biosynthesis pathway by regulating the expression levels of structural genes (*Erika et al., 2015*). Anthocyanin biosynthesis refers to the series of enzymatic reactions that lead to the formation of anthocyanin pigments from the precursor phenylalanine in plants (*Sharma et al., 2024*). These TFs can modulate anthocyanin synthesis either independently or by forming complexes, such as the ternary complex composed of an R2R3-MYB TF, a basic helix-loop-helix (bHLH) domain protein and a WD-repeat protein, collectively known as the MYB-bHLH-WD40 (MBW) complex (*An et al., 2012*; *Liu et al., 2024a*; *Xie et al., 2012*). MYB proteins play significant roles in regulating the biosynthesis of anthocyanin and flavonol compound in plants. A total of 137 R2R3-MYB TFs have been identified in *A. thaliana*. Among these, AtPAP1 (AtMYB75), AtPAP2 (AtMYB90), AtPAP3 (AtMYB113), and AtPAP4 (AtMYB114) have been confirmed to be involved in anthocyanin synthesis (*Borevitz et al., 2000*; *Heppel et al., 2013*; *Nesi et al., 2001*; *Ramsay & Glover, 2005*; *Stracke et al., 2007*), while AtMYB123 regulates proanthocyanidin biosynthesis (*Lepiniec et al., 2006*). Furthermore, the over-expression of *PpMYB108* significantly enhances anthocyanin biosynthesis in tobacco flowers and interacts with the *PpDFR* promoter in peach (*Khan et al., 2022*). PpMYB15 and PpMYBF1 are implicated in the regulation of flavonol biosynthesis in peach fruit (*Cao et al., 2019*). ZeMYB32, identified as an R2R3-MYB, negatively regulates anthocyanin biosynthesis in *Zinnia elegans* (*Jiang et al., 2024*). Additionally, the over-expression of *MYB10.1* in tobacco modulates reproductive anthocyanin biosynthesis (*Rahim et al., 2019*). PbMYB1L in the peel of 'Red Zaosu' pear acts as a positive regulator of anthocyanin biosynthesis (*Zhou et al., 2024*). However, the functions of MYB proteins related to anthocyanin synthesis in 'Sahong Tao' remain unknown.

The yeast one-hybrid (Y1H) technique is a molecular biology method employed to investigate protein-DNA interactions involving specific sequences. (*Berenson et al., 2023*; *Sun et al., 2017*). In lilies, the Y1H assay was utilized to screen ethylene response factors (ERFs), auxin/indole-3-acetic acid (AUX/IAA), and basic transcription factor 3 (BTF3) proteins that are involved in regulating anthocyanin biosynthesis (*Yuwei et al., 2021*). In Qingke (*Hordeum vulgare* L. var. *Nudum* Hook. f), HvANT2 exhibited promoter-binding activity towards anthocyanin-related genes such as *CHI*, dihydroflavonol-3′-hydrogenase (*F3′H*), and *UDP-glucosyltransferase* (*GT*) (*Wang et al., 2024*). In blueberries, VcMYB-1 and VcAN1 have been shown to bind to the *VcGSTF8* promoter, thereby regulating anthocyanin biosynthesis (*Zhang et al., 2024b*). DcMYB11c binds to the promoters of *DcUCGXT1* and *DcSAT1*, contributing to anthocyanin accumulation in carrot purple petioles (*Duan et al., 2023*). In apples, MdNAC33 activates the expression of *MdbHLH3*, *MdDFR*, and *MdANS* genes. Furthermore, MdNAC33 was found to interact with MdMYB1, a positive regulator of anthocyanin biosynthesis and accumulation (*Zhang et al., 2024a*).

The synthesis and accumulation of anthocyanins are essential for the formation of flower variegation in 'Sahong Tao'. *LDOX* is a key gene that controls the biosynthesis of pigmented

anthocyanins. However, the mechanisms underlying its regulatory network remain unclear. In this study, we identified MYB TFs that interact with the *LDOX* promoterusing the Y1H assay and next-generation sequencing (NGS) technology. The candidate MYB members were subsequently cloned and their interactions with the *LDOX* promoter were verified. These findings provide a foundation for a deeper understanding of the regulatory pathways involved in anthocyanin biosynthesis in peach.

## MATERIALS & METHODS

### Plant materials

The plant materials utilized in this study were derived from the peach cultivar 'Sahong Tao', which was cultivated under natural conditions in the experimental field of Huaiyin Institute of Technology (119°2′27″E, 33°33′13″N). Red and variegated flower buds were collected during the initial balloon stage. The variegated petals (petals-V), red petals (petals-R), leaves, red sepals (sepals-R), green sepals (sepals-G), pistils, and stamens were collected individually into centrifuge tubes. The samples were promptly frozen in liquid nitrogen and stored at −80 °C until further use.

### Yeast one-hybrid screening and Next-generation sequencing technology analysis

The 'Sahong Tao' Y1H cDNA library and the *LDOX* promoter bait vector (pHIS2-LDOX-QDZ) have been constructed in our laboratory (*Wu et al., 2024*). Y1H screening assays were performed using the Clontech Yeast One-Hybrid System, provided by ProNet Biotech Co., Ltd (Nanjing, China). All positive clones from the yeast sieve plate were scraped into 2× YPDA liquid, then centrifuged and collected. Colony PCR was performed, and the PCR products were sequenced using the Illumina NovaSeq 6000 platform (ProNet Biotech Co., Ltd, Nanjing, China), with a typical fragment length of 150 bp. Gene IDs, CDS and protein sequences were obtained by performing a BLAST search against the peach genome database downloaded from NCBI (https://ftp.ncbi.nlm.nih.gov/genomes/all/GCF/000/346/465/GCF_000346465.2_Prunus_persica_NCBIv2/). The NGS data were aligned with the *Arabidopsis* database (TAIR10) using Python 3.10.4 software, applying an e-value threshold of less than $1 \times 10^{-10}$ to identify homologous genes. PlantTFDB version 5.0 (https://planttfdb.gao-lab.org/prediction.php) was used to determine whether the proteins were TFs (Supplementary Data S2). Additionally, PlantRegMap (https://plantregmap.gao-lab.org/binding_site_prediction.php) was used to predict TFs that interact with the *LDOX* promoter (Supplementary Data S3).

### Candidate MYB members' gene cloning

Total RNA was isolated from petals-V, petals-R, leaves, sepals-R, sepals-G, pistils, and stamens using the FastPure® Universal Plant Total RNA Isolation Kit (RC411, Vazyme, Nanjing, China). Then, the cDNA was synthesized using the Goldenstar™ RT6 cDNA Synthesis Kit (TSK301S, Tsingke, Beijing, China). Based on the results of NGS and predictions obtained from the PlantRegMap website, MYB44 (PRUPE_6G229000 and PRUPE_1G430000) and MYB1R1 (PRUPE_5G182000) were identified as candidate MYB
**Table 1 The primer sequences used in this study.**

| Primer name | Primer sequence (5′-3′) | Application |
| --- | --- | --- |
| PRUPE_5G182000-C-F | ATGGCTGGCACGTGCTC | CDS full-length clone |
| PRUPE_5G182000-C-R | TCAAGCAACACTAATAATGCTATCCC | CDS full-length clone |
| PRUPE_1G430000-C-F | ATGGCTTCCACAAAGAA | CDS full-length clone |
| PRUPE_1G430000-C-R | CTACTCGATTTTGCTAATCC | CDS full-length clone |
| PRUPE_6G229000-C-F | ATGGCAGTGAGCAGAAAAGA | CDS full-length clone |
| PRUPE_6G229000-C-R | TCACTCGATCCTGCTAACAC | CDS full-length clone |
| pGADT7-F | TAATACGACTCACTATAGG | Yeast one-hybrid assay |
| pGADT7-R | GGCAAAACGATGTATAAATGA | Yeast one-hybrid assay |
| pHIS2-F | TTCGCTATTACGCCAGCTG | Yeast one-hybrid assay |
| pHIS2-R | GTTTATCTTGCCTGCTCATT | Yeast one-hybrid assay |
| PRUPE_5G182000-AD-F | GCCATGGAGGCCAGTGAATTCAT GGCTGGCACGTGCTC | Yeast one-hybrid assay |
| PRUPE_5G182000-AD-R | cagctcgagctcgatggatccTCAAGCAAC ACTAATAATGCTATCCC | Yeast one-hybrid assay |
| PRUPE_1G430000-AD-F | GCCATGGAGGCCAGTGAATTCATG GCTTCCACAAAGAA | Yeast one-hybrid assay |
| PRUPE_1G430000-AD-R | cagctcgagctcgatggatccCTACTCGA TTTTGCTAATCC | Yeast one-hybrid assay |
| PRUPE_6G229000-AD-F | GCCATGGAGGCCAGTGAATTCATG GCAGTGAGCAGAAAAGA | Yeast one-hybrid assay |
| PRUPE_6G229000-AD-R | cagctcgagctcgatggatccTCACTCGAT CCTGCTAACAC | Yeast one-hybrid assay |
| PRUPE_5G182000-E-F | ACCTTGGGACACGCAAATCT | qRT-PCR |
| PRUPE_5G182000-E-R | GATGACGATTCCCTCGGGTC | qRT-PCR |
| RPII-F | TGAAGCATACACCTATGATGATGAAG | qRT-PCR |
| RPII-R | CTTTGACAGCACCAGTAGATTCC | qRT-PCR |
| LDOX-F | GATGCAGGGAGGAGTTGAAG | qRT-PCR |
| LDOX-R | CTGCCCAGAAGCATTGTTTG | qRT-PCR |

members that may interact with the *LDOX* promoter. Using cDNA as template, Primer Premier 6.0 software was employed to design primers (Table 1) for cloning the full-length CDS sequences. The CDS sequences were cloned using the GoldenStar® T6 Super PCR Mix Ver.2 (1.1×) (TSE102, Tsingke, Beijing, China). PCR products were separated by 1% agarose gel electrophoresis, and the gel was observed and cut under using a multifunctional gel image analysis system (Tanon-1160, Tanon, Shanghai, China). Gel strips were purified according to the instructions of DNA Gel Extraction Kit (TSP601-50, Tsingke, Beijing, China). The target fragment was introduced into the pClone007 Blunt vector following the kit instructions (TSV-007B, Tsingke, Beijing, China). The ligation products were transformed into *Escherichia coli* DH5α competent cells (TSC-C01, Tsingke, Beijing, China), and positive clones were sent to Beijing Tsingke Biotech Co., Ltd. (Beijing, China) for sequencing.

## Verification of *LDOX* promoter interaction with MYB

Y1H assay was performed to verify the interaction between the *LDOX* promoter and the candidate MYB members. The *LDOX* promoter sequence was constructed into the

pHIS2 vector, while the CDS of candidate MYB members were inserted into the pGADT7 vector. Subsequently, the constructed vectors were transformed into the yeast strain Y187 through co-transfection and cultured on SD-TL plates. The experimental groups included pHIS2-LDOX-1+pGADT7-PRUPE_5G182000, pHIS2-LDOX-1+pGADT7-PRUPE_1G430000, and pHIS2-LDOX-1+pGADT7-PRUPE_6G229000. The control group was pHIS2-LDOX-1+pGADT7, while positive control group was pGAD53m+pHIS2-p53. The experimental, control and positive control groups were inoculated on SD-TL, SD-TLH, and SD-TLH+50 mM 3AT medium 30 °C constant temperature for 3–5 days. The interaction was confirmed based on the growth status of the yeast cells.

## The predictive analysis of the MYB1R1-*LDOX* promoter interaction relationship

The 3D structures of MYB1R1 protein and *LDOX* promoter were constructed using AlphaFold2, with pLDDT values ranging from 0 to 100 to estimate prediction confidence (*Jumper et al., 2021*). The HDOCK (http://hdock.phys.hust.edu.cn/) was employed to obtain a confidence score, which indicates the likelihood of binding between two molecules. A confidence score exceeding 0.7 suggests a high probability of binding (*Huang & Zou, 2014*). Finally, PyMOL 3.1.4.1 software and PDBePISA (https://www.ebi.ac.uk/msd-srv/prot_int/pistart.html) were used to predict the binding sites of the corresponding protein sequences.

## Bioinformatics analysis of MYB1R1

The *MYB1R1* gene clone and sequencing information was utilized to perform a comprehensive bioinformatics analysis. The molecular weight, protein isoelectric point, instability index, aliphatic index, and grand average of hydropathicity were assessed using Expasy-ProtParam (https://web.expasy.org/protparam/). SignalP 4.1 (https://services.healthtech.dtu.dk/services/SignalP-4.1/) was utilized to forecast the signal peptide. Membrane structure analysis was conducted with TMHMM Server v2.0 (https://services.healthtech.dtu.dk/services/TMHMM-2.0/). Furthermore, prediction of subcellular localization of plant proteins using Plant-mPLoc (http://www.csbio.sjtu.edu.cn/bioinf/plant-multi/), and phosphorylation site analysis was performed using NetPhos 3.1 (https://services.healthtech.dtu.dk/services/NetPhos-3.1/). The secondary and tertiary structures of the protein were predicted using SOPMA (https://npsa.lyon.inserm.fr/cgi-bin/npsa_automat.pl?page=/NPSA/npsa_sopma.html) and SWISS-MODEL (https://swissmodel.expasy.org/interactive), respectively. Protein interaction analysis was carried out utilizing STRING (https://cn.string-db.org/).

## Phylogenetic analysis and alignment of the deduced amino acid sequence of MYB1R1

The amino acid sequence of MYB1R1 was obtained from the NCBI database. Phylogenetic tree construction was carried out using MEGA 11 software, employing the neighbor-joining method (*Tao et al., 2019*). This phylogenetic analysis included homologous amino acid sequences of MYB1R1 from various plants, with peach serving as a reference sequence. The conserved domains were predicted and analyzed using the SMART (https://smart.embl.de/).

Multiple sequence alignment and functional domain labeling were performed using DNAMAN V6 software.

### MYB1R1 promoter analysis

MethPrimer (http://www.urogene.org/cgi-bin/methprimer/methprimer.cgi) was employed to identify CpG islands, criteria used: island size > 100, GC percent > 40.0, Obs/Exp > 0.6, while PlantCARE (http://bioinformatics.psb.ugent.be/webtools/plantcare/html/) was used to analyze the *cis*-acting elements within the 2 kb promoter sequence located upstream of the start codon of the *MYB1R1* gene.

### Analysis of the expression pattern of the *MYB1R1* and *LDOX*

The expression profiles of *MYB1R1* and *LDOX* in various tissues were detected and analyzed *via* qRT-PCR with ChamQ Blue Universal SYBR qPCR Master Mix (Q312, Vazyme, Nanjing, China). The total reaction volume was 20 μL, comprising 10 μL of 2× ChamQ Blue Universal SYBR qPCR Master Mix, one μL of cDNA template, 0.4 μL of forward primer, 0.4 μL of reverse primer, and ddH$_2$O to reach a final volume of 20 μL. The primer sequences are as listed in Table 1. Amplification was performed using the Bio-Rad CFX96 Real-Time PCR system (BIO-RAD, Hercules, CA, USA) with the following program: 1 cycle at 95 °C for 10 s, followed by 40 cycles at 95 °C for 5 s and 60 °C for 30 s. Dissolution curves were generated under the following conditions: 95 °C for 30 s, 95 °C for 10 s, 60 °C for 30 s, 70 °C for 2 s, and 95 °C for 2 s. *RPII* was utilized as the internal reference gene (*Yan et al., 2012*). The relative expression of *MYB1R1* and *LDOX* was calculated using the $2^{-\Delta\Delta Ct}$ method (*Livak & Schmittgen, 2001*).

## RESULTS

### Screening out regulatory transcription factors interaction with *LDOX* promoter

Based on the Y1H assay and the results from NGS of positive clones, a total of 1,190 proteins were identified as interacting with the *LDOX* promoter (Data S1). The NGS data were compared using BLAST against the *Arabidopsis* database (TAIR10), resulting in the identification of 1,146 genes exhibiting high similarity to *Arabidopsis*. Using PlantRegMap, we predicted that 324 TFs bind to the *LDOX* promoter (Fig. 1A). Among the 1,190 proteins, 20 were categorized as TFs, which included four ERF (PRUPE_5G117800, PRUPE_4G176200, PRUPE_4G222300, PRUPE_3G032300), three MYB (PRUPE_2G192100, PRUPE_6G229000, PRUPE_1G430000), two NF-YB (PRUPE_6G130500, PRUPE_5G214400), two SBP (PRUPE_4G050400, PRUPE_1G224900), one S1Fa-like (PRUPE_6G239600), one TCP (PRUPE_5G088800), one bHLH (PRUPE_4G201500), one LBD (PRUPE_4G105700), one ZF-HD (PRUPE_3G267900), one C3H (PRUPE_3G080200), one DBB (PRUPE_1G398700), one MYB_related (PRUPE_5G182000), and one HD-ZIP (PRUPE_6G193400) (Fig. 1B). Of the total 1190 proteins analyzed, 332 were classified as RNA-binding proteins, while 124 were classified as DNA-binding proteins. Additionally, 13 proteins were found to belong to both RNA-binding and DNA-binding proteins (Fig. 1C). Comparison with the

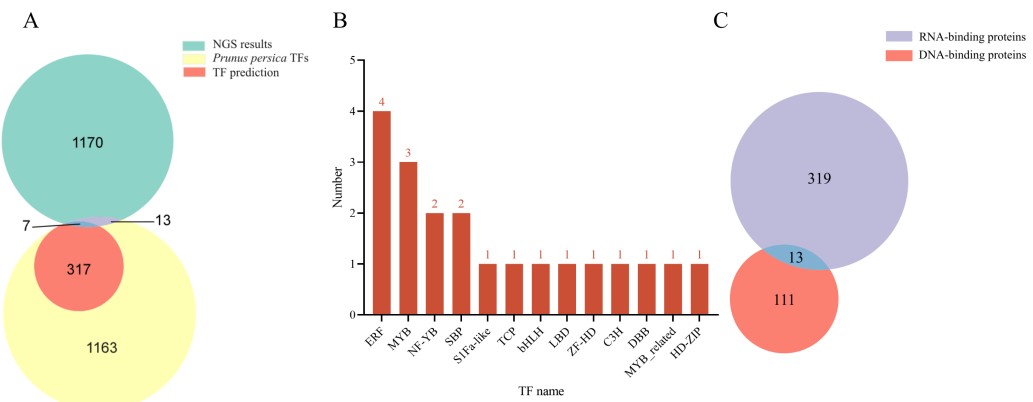

**Figure 1 Analysis of the TFs screened out by NGS.** (A) Venn diagram comparing NGS results with *Prunus persica* TFs and predicted TFs for binding to *LDOX* promoter. (B) Distribution map illustrating the TFs with in each family. (C) Identification results for DNA-binding proteins and RNA-binding proteins.

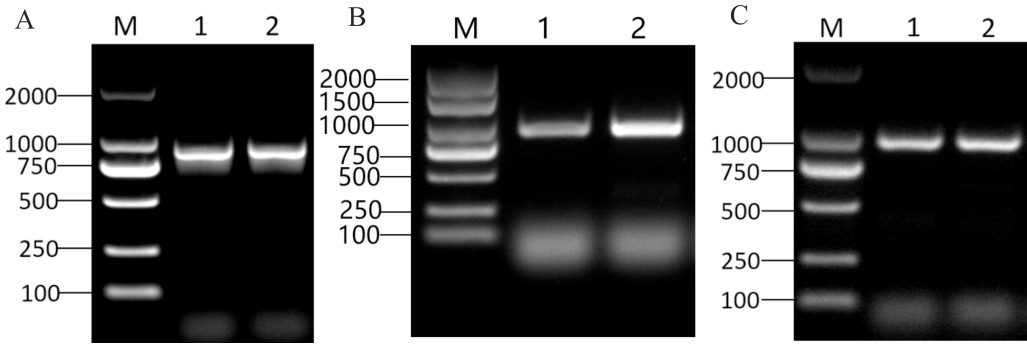

**Figure 2 Cloning of candidate MYB TFs.** M: DNA marker DL2000. (A) 1–2: CDS clone of the PRUPE_5G182000, (B) 1–2: CDS clone of the PRUPE_1G430000, (C) 1–2: CDS clone of the PRUPE_6G229000.

NGS results identified seven TFs that were consistent with PlantRegMap predictions, including three ERF (PRUPE_4G176200, PRUPE_4G222300, PRUPE_3G032300), one HD-ZIP (PRUPE_6G193400), one MYB_related (PRUPE_5G182000), two MYB (PRUPE_6G229000, and PRUPE_1G430000).

## Candidate MYB members' gene cloning

The CDS sizes for PRUPE_5G182000, PRUPE_1G430000, and PRUPE_6G229000, were 897 bp (Fig. 2A), 939 bp (Fig. 2B), and 996 bp (Fig. 2C), respectively. Sanger sequencing results of the positive clone colonies aligned with the analysis of NGS results conducted using DNAMAN V6 software. The correct CDS were then employed to construct the prey vectors pGADT7-PRUPE_6G229000, pGADT7-PRUPE_1G430000, and pGADT7-PRUPE_5G182000, which were used to verify their interactions with the *LDOX* promoter.

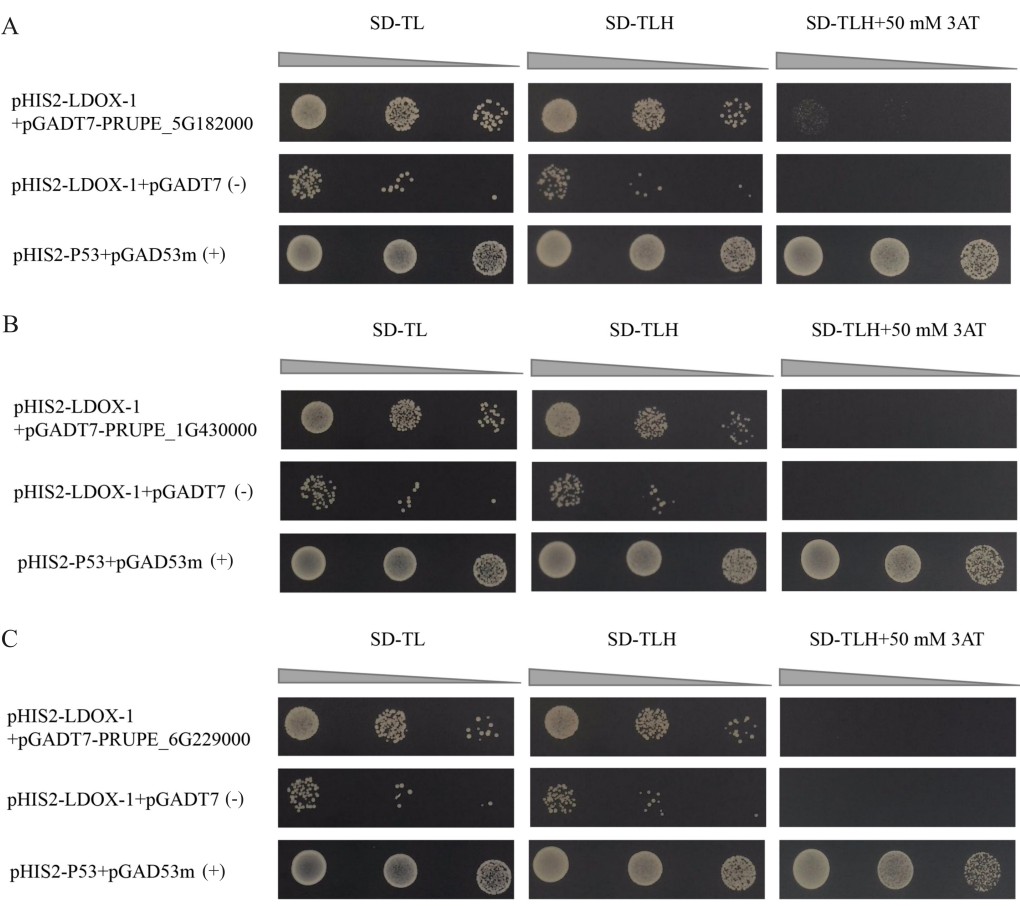

**Figure 3 Interaction validation results of pGADT7-PRUPE_5G182000 (A), pGADT7-PRUPE_1G430000 (B), and pGADT7-PRUPE_6G229000 (C) with pHIS2-LDOX-1.** (+): pHIS2-P53+pGAD53m as positive control, (-): pHIS2-LDOX-1+pGADT7 as negative control. SD-TL: -trp, -leu; SD-LH: -leu, -his; SD-TLH: -trp, -leu, -his. 3AT: a competitive inhibitor of yeast HIS3 protein synthesis, used to inhibit leaky expression of the *HIS3* gene.

## Identification of MYB1R1 as a potential regulator of LDOX

The results indicated that the positive control exhibited normal growth (pGAD53m+pHIS2-p53) on SD-TL, SD-TLH, and SD-TLH+50 mM 3AT plates. In the control group (pHIS2-LDOX-1+pGADT7), normal growth was observed on SD-TL and SD-TLH plates. However, no growth occurred on SD-TLH+50 mM 3AT plates. The combination of pHIS2-LDOX-1+pGADT7-PRUPE_5G182000 displayed weak growth observed on SD-TLH+50 mM 3AT plates, indicating positive growth (Fig. 3A). Conversely, pHIS2-LDOX-1+pGADT7-PRUPE_1G430000 (Fig. 3B) and pHIS2-LDOX-1+pGADT7-PRUPE_6G229000 (Fig. 3C) did not exhibit growth on SD-TLH+50 mM 3AT plates. These results demonstrate that pGADT7-PRUPE_5G182000 interacts with pHIS2-LDOX-1.

## Y1H-AOS predicted analysis

Y1H-AOS stands for analysis of confidence in nucleic acid-protein interactions. Y1H-AOS models the structure of the *LDOX* promoter and the MYB1R1 protein through homology

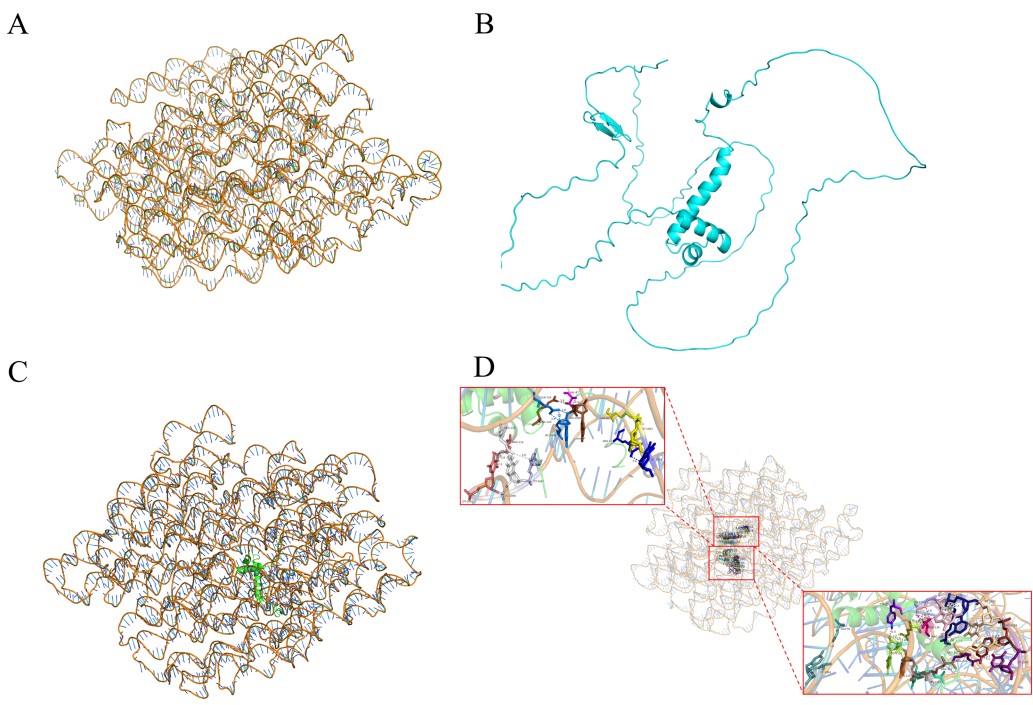

**Figure 4** **MYB1R1 predictive analysis.** (A) 3D structure of *LDOX* promoter. (B) 3D structure of MYB1R1 protein. (C) 3D structure of two sequences docking. (D) Analysis of the binding-site.

modeling to predict their interaction. The predominant elements of the 3D structure of the *LDOX* promoter exhibit a double helical structure (Fig. 4A), while the MYB1R1 protein includes $\alpha$-helix and irregular coil (Fig. 4B). The pLDDT value obtained from AlphaFold2 was 60.63, and the confidence score from HDOCK was 0.9763, indicating the overall high reliability and stability of this nucleic acid protein composite structure. Although the pLDDT values from AlphaFold2 suggest some local uncertainty or flexibility in certain regions, the high confidence values from HDOCK further support the overall stability and quality of this composite structure. The analysis identified 24 binding sites (Fig. 4C; Table 2) where MYB1R1 protein amino acid residues interact with the *LDOX* promoter to form hydrogen bonds, with no disulfide, covalent, or salt bridges detected at these binding sites (Fig. 4D).

## Analysis of MYB1R1 protein properties

In this study, we examined the *MYB1R1* gene, which encodes a protein consisting of 298 amino acids (Fig. 5A). The MYB1R1 protein has a molecular weight of 32.49 kDa, with a molecular formula of $C_{1392}H_{2219}N_{425}O_{450}S_{12}$, corresponding to a total atomic number of 4498. Theoretical isoelectric point value of 7.20 inferred that it is a basic protein. The minimum hydrophilic value ($-4.022$) was predicted at amino acid sites 20 and 21, while the maximum hydrophilic value (2.300) was found at sites 139, 140, and 141. The average total hydrophilic coefficient was calculated to be $-0.599$, indicating that this protein is hydrophilic (Fig. 5B). Additionally, prediction results for the protein signal

**Table 2   Hydrogen bonds between the *LDOX* promoter and MYB1R1 protein.** "Dist" represents distance. Each row indicates the binding sites between amino acid residues of the protein and nucleotide positions in the nucleic acid, along with their specific distance measurements. "Hydrogen bonds" denotes the hydrogen bonding interactions between them.

| Number | MYB1R1 | Dist [Å] | *LDOX* promoter |
|---|---|---|---|
| 1 | A:ARG 81[NH1] | 3.55 | B:DT1053[O4] |
| 2 | A:ARG 81[NH2] | 3.67 | B:DT1053[O2] |
| 3 | A:ARG 81[NH2] | 2.57 | B:DT1054[O2] |
| 4 | A:ARG 83[NH2] | 3.61 | B:DT1053[O4] |
| 5 | A:VAL 87[N] | 2.37 | B:DG1050[OP1] |
| 6 | A:ARG 110[NH1] | 2.54 | B:DT1045[O4] |
| 7 | A:ARG 110[NH2] | 3.86 | B:DT1047[O4] |
| 8 | A:ARG 110[NH2] | 2.32 | B:DC1046[N3] |
| 9 | A:ARG 114[NH1] | 2.33 | B:DT1045[O2] |
| 10 | A:ARG 120[NH1] | 3.75 | B:DG1050[OP1] |
| 11 | A:GLN 124[NE2] | 2.80 | B:DC1049[O3′] |
| 12 | A:TYR 132[OH] | 2.37 | B:DG1154[O3′] |
| 13 | A:ARG 136[NH1] | 2.83 | B:DG1154[O3′] |
| 14 | A:ASN 138[ND2] | 2.84 | B:DG1180[O5′] |
| 15 | A:ARG 141[NH1] | 3.87 | B:DG1180[O3′] |
| 16 | A:ARG 143[NH2] | 2.75 | B:DT1152[O3′] |
| 17 | A:ARG 143[NH2] | 2.01 | B:DA1153[OP1] |
| 18 | A:ARG 144[NH1] | 3.32 | B:DT1184[O4] |
| 19 | A:ARG 145[NH1] | 3.06 | B:DT1182[O2] |
| 20 | A:ARG 145[NH2] | 3.80 | B:DG1183[O3′] |
| 21 | A:ARG 145[O] | 2.78 | B:DA1181[N1] |
| 22 | A:ARG 141[O] | 3.04 | B:DA1181[N6] |
| 23 | A:ARG 145[O] | 2.84 | B:DA1181[N6] |
| 24 | A:GLU 91[OE1] | 3.81 | B:DC1861[N4] |

peptide indicate that the MYB1R1 protein does not possess a signal peptide (Fig. 5C). The predicted number of transmembrane structures for MYB1R1 is 0, confirming that it lacks any transmembrane structure (Fig. 5D). Predict subcellular localization of MYB1R1 protein in the nucleus (Fig. 5E). Phosphorylation is predicted to occur within the MYB1R1 peptide chain, with 35 amino acid sites exhibiting scores above 0.5. This includes 21 serine, 12 threonine, and 2 thyronine phosphorylation sites (Fig. 5F).

## MYB1R1 protein structure prediction analysis

The MYB1R1 protein secondary structure prediction results indicated that its composition primarily consists of α-helix (13.76%), β-helix (2.35%), random coil (68.12%), and extended chain (15.77%) (Fig. 6A). The predicated tertiary structure exhibited a GMQE value of 0.56 and a similarity value of 97.32% (Fig. 6B). Notably, both the secondary and tertiary structure prediction results were consistent, with random coil identified as the predominant structural component of the MYB1R1 protein.

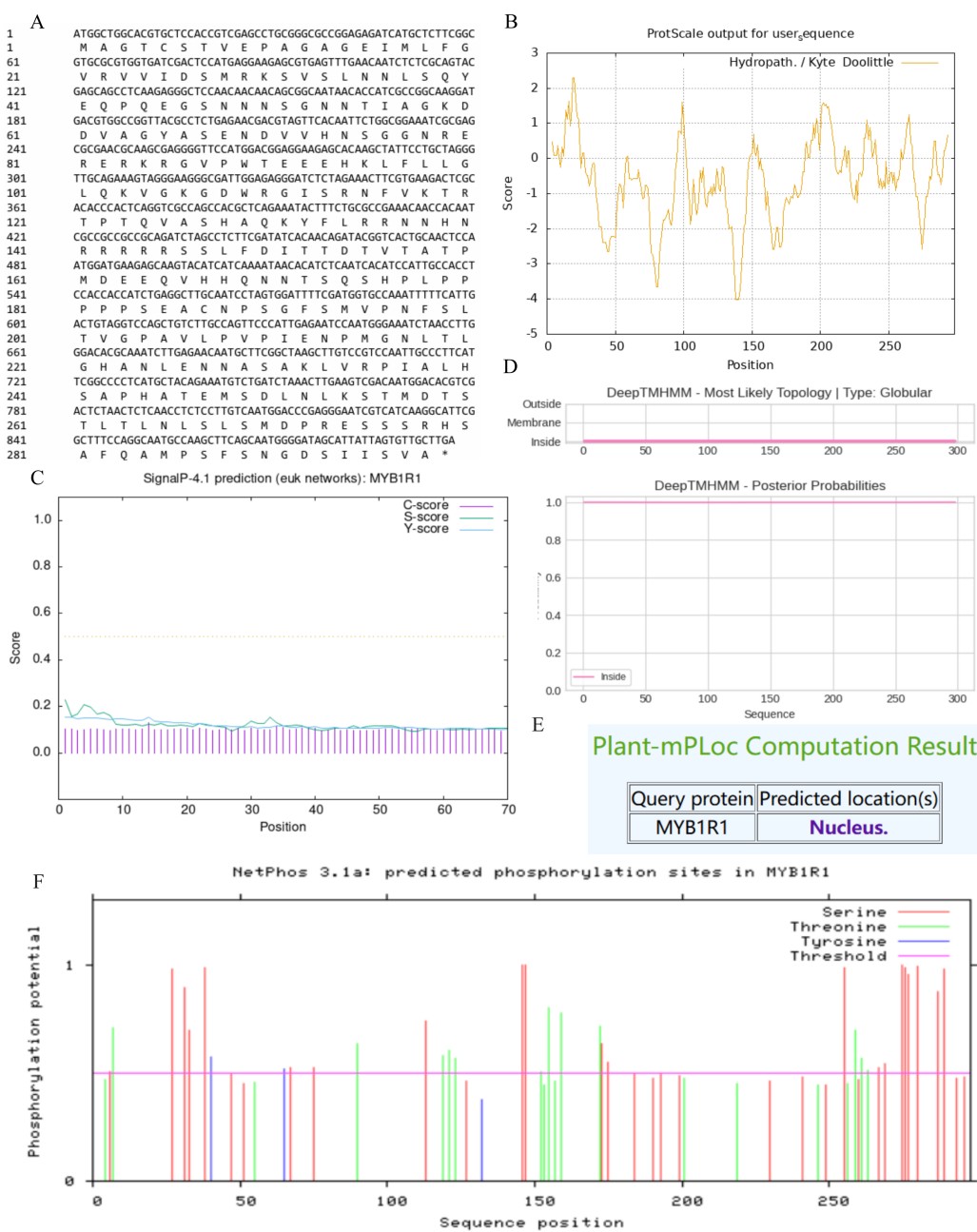

**Figure 5 MYB1R1 protein physicochemical properties analysis.** (A) Amino acid sequence. (B) Prediction of hydrophilic/hydrophobicity. (C) Prediction of signal peptide. (D) Prediction of transmembrane structure. (E) Prediction of the subcellular localization. (F) Prediction of phosphorylation site.

## Analysis of the interaction proteins of MYB1R1

Using STRING to predict the interacting proteins of MYB1R1, the results showed that MYB1R1 may interact with ten proteins (Fig. 7), including bHLH domain-containing proteins (PRUPE_6G211900, PRUPE_6G212000, and PRUPE_1G434000), NAC TFs (NAC1 and NAC2), homeobox domain-containing protein (PRUPE_6G080100),

A

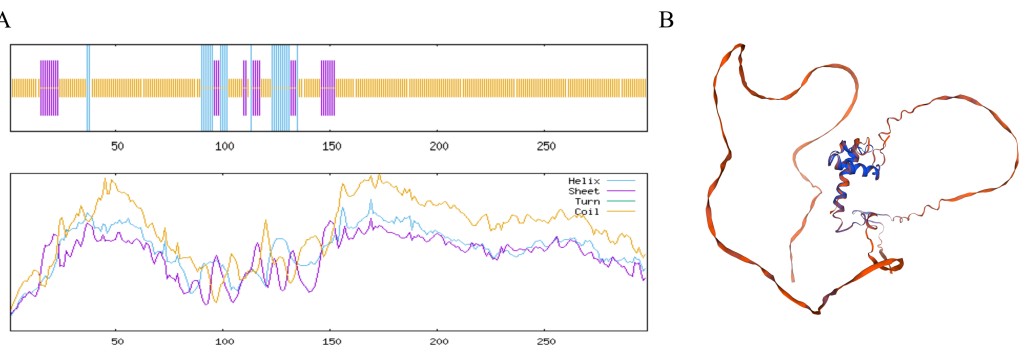

B

**Figure 6 Structure prediction of the MYB1R1 protein.** (A) Secondary structure prediction of the MYB1R1 protein. (B) Tertiary structure prediction of the MYB1R1 protein.

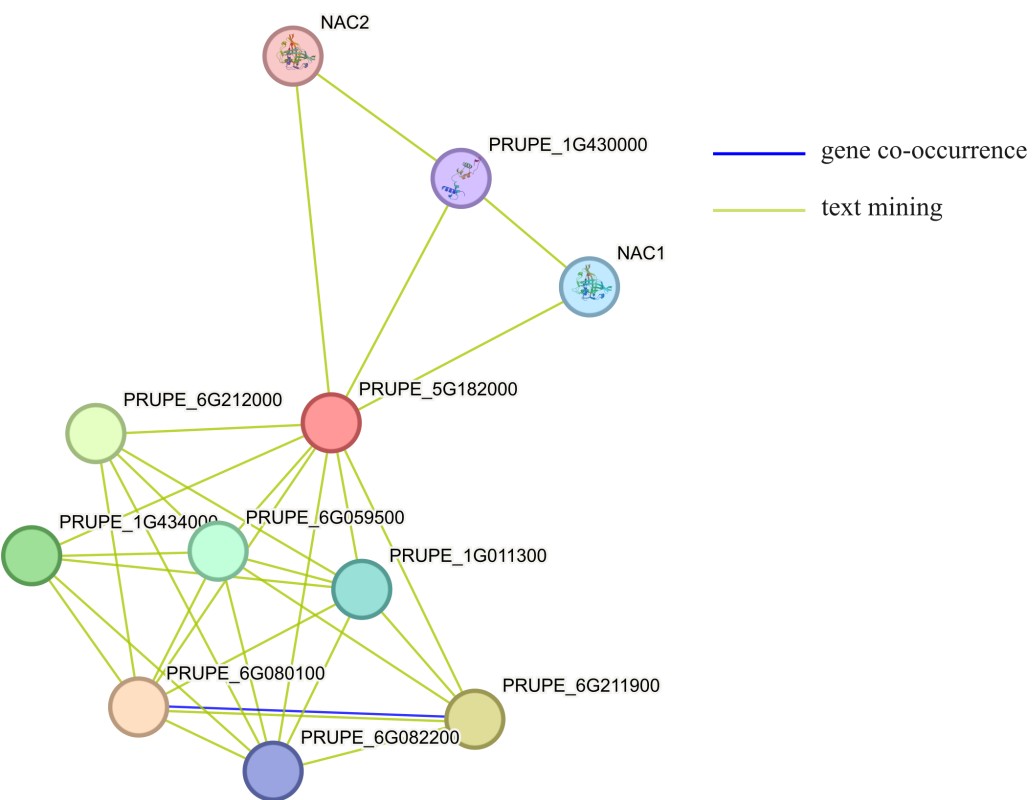

**Figure 7 The protein–protein interaction network analysis of MYB1R1.** The circles, depicted in various colors, represent the proteins that interact with MYB1R1. Different line colors denote various active interaction sources: blue lines indicate gene co-occurrence, while yellow lines represent text mining.

cell division control protein (PRUPE_6G059500), ADF-H domain-containing protein (PRUPE_6G082200) and uncharacterized proteins (PRUPE_1G430000 and PRUPE_1G011300). This suggests that MYB1R1 may interact with additional proteins to perform various functions.

### Sequence alignment and phylogenetic analysis of *MYB1R1* gene

After aligning the amino acid sequence encoded by the *MYB1R1* gene from NCBI, we performed a homology sequence alignment analysis using DNAMAN V6 software (Fig. 8A). This analysis revealed a high degree of homology (>70%) with the amino acid sequences of *P. avium*, *P. speciosa*, *Pyrus communis*, *Pyrus × bretschneideri*, *Rosa chinensis*, *Rosa rugosa*, *Quercus robur*, and *Morella rubra*. The evolutionary tree indicates that the protein encoded by the *MYB1R1* gene has the closest phylogenetic relationship with *P. avium* and *P. speciosa* (Fig. 8B). Additionally, prediction and analysis of conserved domains using the SMART online tool demonstrated that the *MYB1R1* gene-encoded protein shares the SANT-MYB domain with the aligned sequences from other species.

### Prediction of CpG island and *cis*-acting elements in *MYB1R1* promoter

The results of CpG island prediction for the promoter region of MYB1R1 indicate the presence of one CpG island within *MYB1R1*, which is 91 bp in length and located at the position of 471–561 bp (Fig. 9A). The results revealed that the *MYB1R1* promoter contains a total of 32 distinct types of *cis*-acting elements. Apart from the conserved promoter elements CAAT-box and TATA-box, it includes light-responsive regulatory elements such as Box4, I-box, GATA-motif, ATCT-motif, G-box, TCT-motif, AE-box, and GT1-motif. Additionally, there are hormone-responsive elements like ABRE, TGAGG-motif, CGTCA-motif, AuxRR-core, and TCA-element. The promoter also harbors MYB TF binding elements such as MBS, as well as growth and development regulatory elements including Circadian, GCN4_motif, HD-Zip, and O2-site. Furthermore, MBS and LTR are *cis*-elements involved in stress responsiveness (Fig. 9B). These *cis*-acting elements are likely to influence the ultimate gene expression.

### Gene expression analysis of *MYB1R1* and *LDOX*

The gene expression of *MYB1R1* and *LDOX* in various tissues of 'Sahong Tao' was analyzed using qRT-PCR (Fig. 10). The results showed that the expression level of *MYB1R1* in variegated petals is higher than that in red petals, whereas the expression level of *LDOX* in variegated petals is lower than that in red petals. It is speculated that MYB1R1 interacts with LDOX to negatively regulate the synthesis of anthocyanins. Furthermore, nearly no gene expression was observed in sepals-R and sepals-G.

## DISCUSSION

### Structural genes *LDOX* control the synthesis of anthocyanins

Anthocyanins are a group of naturally water-soluble pigments responsible for color changes in vegetative tissues and reproductive organs (*Sharma et al., 2024*). The color variation in the petals of 'Sahong Tao' is influenced by anthocyanin synthesis, which is regulated by the expression of structural genes encoding key enzymes such as *PAL*, *CHS*, *CHI*, *F3H*, *F3′H*, *F3′5′H*, *DFR*, *LDOX/ANS*, and *UFGT*, as well as TFs (*Ferrer et al., 2008*; *Zohar et al., 2015*). *LDOX* catalyzes the synthesis of unmodified colored anthocyanins from leucoanthocyanidins. During the veraison to ripening stages of grape berries, *LDOX*

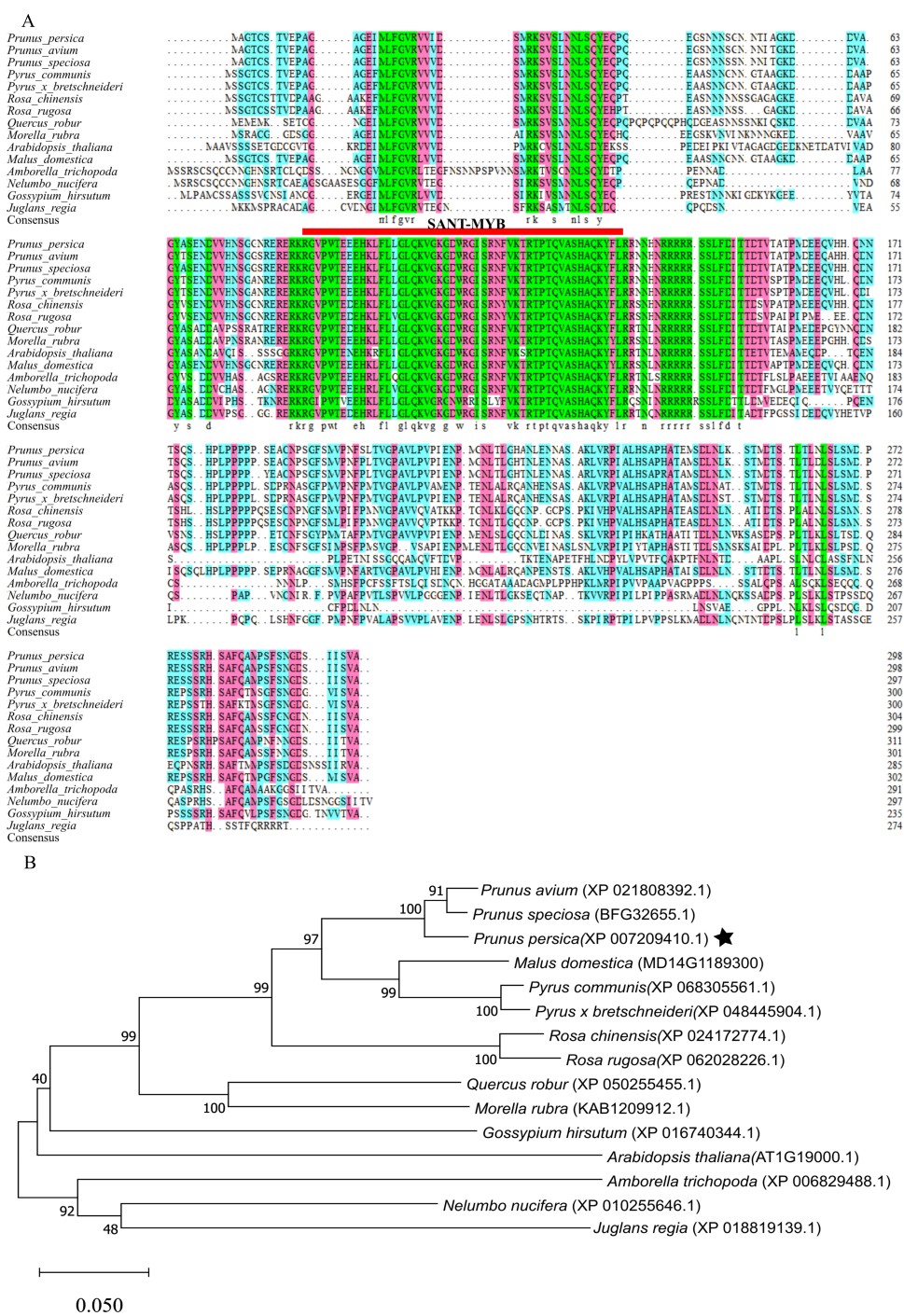

**Figure 8  Multiple sequence alignment and phylogenetic analysis of *MYB1R1* gene.** (A) Multiple sequence alignment of amino acid sequences among MYB1R1 and other plants. (B) Phylogenetic tree analysis of MYB1R1 protein.

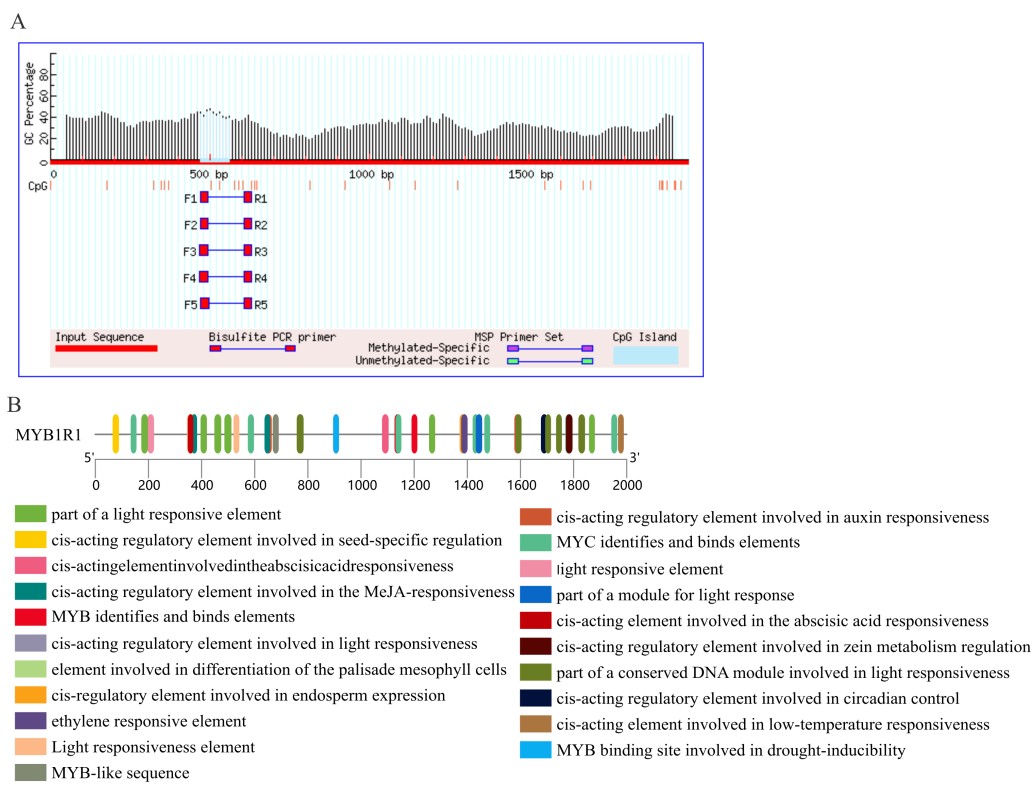

**Figure 9** **Prediction of CpG island and *cis*-acting elements in *MYB1R1* promoter.** (A) Prediction of CpG island in *MYB1R1* promoter. (B) Prediction of *cis*-acting elements in *MYB1R1* promoter.

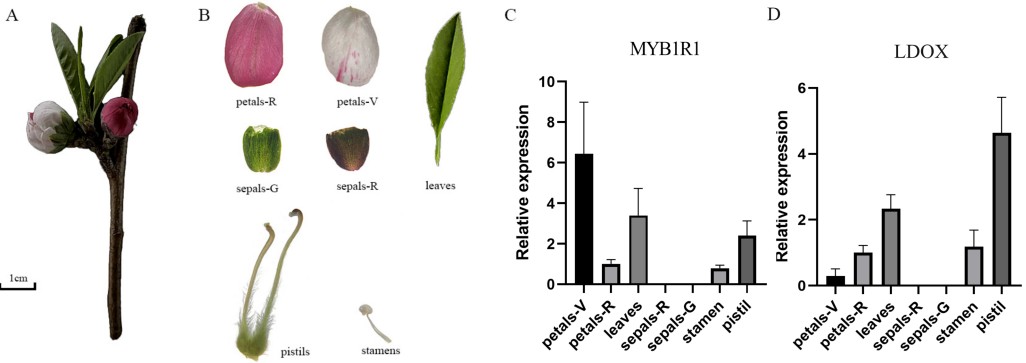

**Figure 10** **Plant materials and the expression analyses of two genes (*MYB1R1* and *LDOX*).** (A) 'Sahong Tao' plant materials used in this study. (B) 'Sahong Tao' pictures of the different part. (C) The relative expression level of *MYB1R1* in different parts of 'Sahong Tao'. (D) The relative expression level of *LDOX* in different parts of 'Sahong Tao'.

exhibits a positive correlation between gene expression and anthocyanin content (*Zhao et al., 2016*). In *Magnolia*, the expression level of *LDOX* genes in red petals is significantly higher than that in white petals (*Shi et al., 2015*). This finding aligns with observations that

both the protein abundance and gene expression of LDOX in the red petals of 'Sahong Tao' are significantly greater than those in white petals (*Zhou et al., 2015a*). Variations in *LDOX* gene expression may affect petal color. However, the regulatory network involving *LDOX* is poorly understood in peach. In this study, NGS and Y1H were used to screen for TFs interacting with the *LDOX* promoter involved in anthocyanin synthesis in the petals of 'Sahong Tao'. This lays a foundation for further revealing the molecular mechanisms of anthocyanin synthesis and accumulation in peaches.

## The role of MYB1R1 in anthocyanin synthesis

TFs play a crucial role in regulating anthocyanin metabolism by interacting with structural genes in plants (*Dubos et al., 2010*). Previously reported MYB TFs, including AtPAP1 (AtMYB75), AtPAP2 (AtMYB90), AtPAP3 (AtMYB113), and AtPAP4 (AtMYB114), as well as the bHLH TT8 and GL3 (*Gao et al., 2017*; *Salez et al., 2022*), and the WRKY (*Shi et al., 2022*; *Wang et al., 2023*), have been shown to be involved in anthocyanin biosynthesis. Among these, MYB proteins represent one of the largest families of TFs in plants (*Hu et al., 2020*). The expression of MYB-6 and *LDOX1* increases under cold stress, leading to the accumulation of major anthocyanins in purple-black carrots (*Dar et al., 2022*). In *Arabidopsis*, the expression levels of *LDOX* in *MYB3* significantly increase under high salinity conditions, resulting in a substantial rise in anthocyanin accumulation (*Kim et al., 2022*). MYB forms the MYB-BHLH-TTG1 complex, which directly regulates the expression of *LDOX*, thereby influencing anthocyanin accumulation in *Arabidopsis* seedlings (*Appelhagen et al., 2011*). PavMYB10.1 is involved in the anthocyanin biosynthesis pathway and determines the skin color of sweet cherries (*Jin et al., 2016*). In peach, the MYB TF *Peace* is highly expressed in pink petals but shows lower expression levels in variegated petals (*Uematsu et al., 2014*). MYB10 and MYBPA1 regulate anthocyanin biosynthesis in the pericarp and proanthocyanidin biosynthesis in the flesh, respectively (*Ravaglia et al., 2013*). Previous studies have identified multiple MYB-related *cis*-acting elements (including MBS, MRE, MYB, and MYB recognition sites) in the *LDOX* promoter of the 'Sahong Tao' (*Wu et al., 2024*). In this study, 1,190 proteins interacting with the *LDOX* promoter were screened out through Y1H and NGS, and three MYB transcription factors were identified. Therefore, MYB TFs and LDOX may be key factors influencing anthocyanin synthesis.

The Y1H analysis is employed to investigate the interactions between proteins and DNA. In eggplant, Y1H has been utilized to substantiate the binding of SmMYB75 to the *SmCHS* promoter facilitating anthocyanin biosynthesis (*Shi et al., 2021*). Furthermore, BoMYBL2b interacts with the MRE site on *ProBoDFR1*, influencing anthocyanin synthesis in kale (*Liu et al., 2024b*). The binding of DcMYB11c to the *DcUCGXT1* and *DcSAT1* promoters is involved in anthocyanin synthesis in purple carrot (*Duan et al., 2023*). Y1H screening and qRT-PCR analysis by ABA signaling demonstrated direct binding of MYC2 and MYB1R1 to the *PbFAD3a* promoter in pears (*Wang et al., 2022*). BLAST comparison of homologous gene in *Arabidopsis* revealed that MYB1R1 is most similar to AT1G19000.2, which has been confirmed to be involved in proanthocyanidin accumulation (*Hong et al., 2017*). It is hypothesized that MYB1R1 may also have a similar function in peaches, and the current results are consistent with previous reports. Therefore, in this study, Y1H was used to

verify the interaction between MYB1R1 and the *LDOX* promoter. Quantitative reverse transcription polymerase chain reaction (qRT-PCR) experiments demonstrated that the expression levels of *MYB1R1* and *LDOX* are inversely correlated in variegated and red flowers of 'Sahong Tao', suggesting that they play a key role in the formation of petal variegation in 'Sahong Tao'. Additionally, MYB1R1 regulates anthocyanin synthesis as a repressive transcription factor.

Bioinformatics analysis of MYB1R1 revealed that the N-terminus of it features a typical SANT-MYB domain. A family of genes containing the SANT/MYB domain has been identified in tomato, where they play a role in regulating plant growth and development (*Barg et al., 2005*). The protein encoded by the *smh1* gene in maize contains a SANT/myb-like domain at its N-terminus and binds to repetitive sequences at the ends of DNA. This leads to the hypothesis that MYB1R1 may similarly bind to repetitive sequences at DNA termini (*Marian & Bass, 2005*). Results from multiple sequence alignment and phylogenetic analysis indicate that the amino acids encoded by *MYB1R1* share close homology with those encoded by genes in plants such as *P. avium*, and *Prunus speciosa*. The MYB in its homologous species has been confirmed to be involved in anthocyanin biosynthesis. In addition to containing conserved promoter *cis*-acting elements such as TATA-box and CAAT-box, the promoter of *MYB1R1* also includes *cis*-acting elements related to light response, hormones, growth and development, and stress response.

## Future directions

DNA methylation is a crucial mechanism of genomic modification (*Bartels et al., 2018*), which can regulate gene expression by interactions with TFs or altering chromosomal structure. Promoter sequences recognized by TFs are prominent in the structure and contain CpG sites (*Liu et al., 2013*). When cytosine at these CpG sites undergoes methylation, TFs sensitive to DNA methylation are no longer able to bind to the corresponding promoter sequence, resulting in the inability to activate and express the associated gene (*Curradi et al., 2002*). In apples and pears, DNA methylation influences anthocyanin accumulation in the peel by regulating MYB expression. The methylation level of *CHS* in orchids determines the accumulation of anthocyanins in flower organs (*Liu et al., 2012*). In this study, MYB1R1 is shown to bind to the *LDOX* promoter to regulate anthocyanin synthesis, leading to petal variegation. In the whole genome methylation sequencing of 'Sahong Tao', the methylation level of the *LDOX* promoter was found to be higher in variegated samples compared to red samples. The methylation sites encompass several regulatory elements related to MYB (*Wu et al., 2020*). However, the difference of *LDOX* promoter methylation level maybe cause the differential expression of *LDOX* in 'Sahong Tao' petals, which ultimately affects anthocyanin synthesis. The effect of *LDOX* promoter methylation on the regulation of MYB1R1 is still unclear. Thus, the role of *LDOX* promoter DNA methylation in this process needs to be further investigated through techniques such as Chromatin Immunoprecipitation (ChIP) and Bisulfite Sequencing.

## CONCLUSIONS

In this study, the Y1H and NGS results indicate that 1,190 proteins interact with the *LDOX* promoter. Among these, 20 TFs were identified. Potential *LDOX* promoter binding TFs were predicted, confirmed that seven TFs matched the Y1H results. We have verified the interaction between MYB1R1 and the *LDOX* promoter and confirmed 24 interaction binding sites by Y1H-AOS. qRT-PCR experiments showed that the expression level of *MYB1R1* in variegated petals is higher than that in red petals and the expression level of *LDOX* in variegated petals is lower than that in red petals. The results showed that MYB1R1 interacted with *LDOX* promoter and negatively regulated the synthesis of anthocyanins. Our findings establish a theoretical foundation for further exploration of the regulatory role of MYB TFs in the flower color of 'Sahong Tao' and the biological functions of *MYB1R1* gene.

## ACKNOWLEDGEMENTS

We thank the Academic Editor and the reviewers for their useful feedback that improved this paper.

### Funding

This research was supported by the National Natural Science Foundation of China (32001358) and College Students' Innovative Entrepreneurial Training Plan Program (202411049289XJ). The funders had no role in study design, data collection and analysis, decision to publish, or preparation of the manuscript.

### Grant Disclosures

The following grant information was disclosed by the authors:
National Natural Science Foundation of China: 32001358.
College Students' Innovative Entrepreneurial Training Plan Program: 202411049289XJ.

### Competing Interests

The authors declare there are no competing interests. Zewen Cao is employed by Beijing Huazhiyuan Horticulture Co., Ltd.

### Author Contributions

- Xinxin Wu conceived and designed the experiments, performed the experiments, analyzed the data, authored or reviewed drafts of the article, and approved the final draft.
- Tong Du conceived and designed the experiments, performed the experiments, analyzed the data, prepared figures and/or tables, authored or reviewed drafts of the article, and approved the final draft.
- Yan Li analyzed the data, authored or reviewed drafts of the article, and approved the final draft.

Peer J

- Weibing Zhuang conceived and designed the experiments, authored or reviewed drafts of the article, and approved the final draft.
- Naixin Kang performed the experiments, prepared figures and/or tables, authored or reviewed drafts of the article, and approved the final draft.
- Jiaxin Zeng analyzed the data, authored or reviewed drafts of the article, and approved the final draft.
- Cong Yan performed the experiments, analyzed the data, authored or reviewed drafts of the article, and approved the final draft.
- Zhenzhu Hu analyzed the data, prepared figures and/or tables, authored or reviewed drafts of the article, and approved the final draft.
- Zewen Cao analyzed the data, authored or reviewed drafts of the article, and approved the final draft.

## Data Availability

The raw data are available at NCBI SRA BioProject: PRJNA1171730.

## Supplemental Information

Supplemental information for this article can be found online at http://dx.doi.org/10.7717/peerj.19975#supplemental-information.

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
