# Peer review of "Screening of MYB1R1 interaction with LDOX promoter to regulate anthocyanin biosynthesis in peaches"

_PeerJ, doi:10.7717/peerj.19975_

## Round 0.1 · original submission · Major Revisions

Please address the concerns of all reviewers and revise the manuscript accordingly.

**Language Note:** The review process has identified that the English language must be improved. PeerJ can provide language editing services - please contact us at [email protected] for pricing (be sure to provide your manuscript number and title). Alternatively, you should make your own arrangements to improve the language quality and provide details in your response letter. – PeerJ Staff

·

Basic reporting

attached

Experimental design

-

Validity of the findings

-

Reviewer 2 ·

Basic reporting

-

Experimental design

-

Validity of the findings

-

Additional comments

I have reviewed the article entitled” Screening of transcription factors interaction with LDOX promoter to regulate the anthocyanin biosynthesis in peach,”. The work seems interesting.
1. The title should be revised.
2. The gene's name should be italicized.
3. Lines No 37-38 should be specific and in line with the current study
4. It is better to improve the language
5. The introduction section should be polished, concise, specific, and revised.
6. Avoid repetition in the Materials and Methods section.
7. In the result section, please precisely and logically explain figures, etc.
8. The discussion section should be in-depth and logically explained.
9. For further details, please have a look at the revised MS article file.

Annotated reviews are not available for download in order to protect the identity of reviewers who chose to remain anonymous.

Reviewer 3 ·

Basic reporting

The authors’ study falls within the scope of the PeerJ. The work has identified 1190 proteins, and among them, MYB1R1 TF was found interacting with the LDOX promoter via Y1H assay. MYB1R1 protein was further characterized. The thesis is well prepared and clearly written. The results obtained use appropriate analytical methods.

Experimental design

Use appropriate analytical methods.

Validity of the findings

-

Additional comments

I have a few comments are need to be addressed in the revised version.

1. The study speculated that MYB1R1 interacts with LDOX to negatively regulate the synthesis of anthocyanins. According to the expression analysis of the MYB1R1 gene via qRT-PCR (Fig. 10), the expression levels of these two genes were opposite, with the expression level of LDOX being lower in striped petals than in red petals. In contrast, the expression level of MYB1R1 was higher in striped petals than in red petals. To strengthen this speculation, the authors should extend the expression analysis of other genes involved in regulating coloration patterns, such as structural genes in the anthocyanin biosynthesis pathway (e.g., PAL, CHS, CHI, F3H, DFR, ANS, UFGT, etc).
2. Fig. 2: Please label notation (A, B, C) as mentioned in its legend for each sub-figure (panel).
3. Fig. 3: It would be better if the authors labeled notation (A, B, C) for the three group vectors [pHIS2-LDOX-1+pGADT7-PRUPE_5G182000; pHIS2-LDOX-1+pGADT7-PRUPE_1G430000; and pHIS2-LDOX-1+pGADT7-PRUPE_6G229000], respectively. This would make the figure more easily observed and tracked between the text and figure data since this figure includes many sub-figures. In addition, please note that the authors presented in this figure’s legend with characters/symbols as “(+), (-)”, denoted for “pHIS2-P53+pGAD53m as positive control” and “pHIS2-LDOX-1+pGADT7, as negative control”. However, no legend notation of characters/symbols [“(+), (-)”] was labeled in Fig. 3.
4. Revise “PLDDT” (L316, 319) to “pLDDT
5. Note that the manuscript includes supplementary data; therefore, please include these data in the Data Availability Statement section.
Best regards,

---

## Round 0.2 · accepted · Accept

All issues pointed out by the reviewers were addressed and the revised manuscript is acceptable now.

Reviewer 2 ·

Basic reporting

The authors have the revised The manuscript properly, and now it can be accepted.

Experimental design

Well-structured and defined

Validity of the findings

The findings are valid

Additional comments

Non

Reviewer 3 ·

Basic reporting

Many thanks for the revisions to your manuscript. The authors have made efforts to improve the manuscript. The authors have now included supplementary data in the Data Availability Statement section as requested. The revised version has been greatly enhanced.

Experimental design

The research question is well defined, with the main objective of this study being to validate the direct interaction between TFs and the LDOX promoter.

Validity of the findings

This research's findings indicate that MYB1R1 interacts with the LDOX promoter, and the qRT-PCR data reveal inverse expression levels of MYB1R1 and LDOX between variegated and red petals. The findings align with the research hypothesis and are supported by the data presented.

Additional comments

I have only one remark for the revised version:
Please provide the accession number/GenBank of genes listed in Table 1, based on which the primer sequences were designed.